# The landscape of hereditary haemochromatosis risk and diagnosis across the British Isles and Ireland

Shona M. Kerr [1,2], Benjamin S. Fletcher [1], Gannie Tzoneva [3], Alan R. Shuldiner [3], Edmund Gilbert [4,5] & James F. Wilson [1,2,6] ✉

Hereditary haemochromatosis is caused by pathogenic variants in the homoeostatic iron regulator gene *HFE*. Outcomes include liver cancer, cirrhosis and arthropathy, but penetrance is incomplete. Here, we use genetic data from >400,000 subjects to determine the genetic risk across 29 regions of the British Isles and Ireland. Northwest Irish and Outer Hebrideans are at the highest risk (1/54 – 1/62 carry the major risk genotype), Mainland Scots are also at increased risk (1/117), declining to 1/212 in Southern England. We also assessed the prevalence of clinically diagnosed haemochromatosis in >63 million people in NHS England and identified 70,365 cases. White Irish individuals have the highest prevalence (3.7x white British). Among white British, prevalence varied 11-fold from 1/1972 in parts of Kent to 1/177 in Liverpool. Discrepancies between genetic risks and prevalences of clinical diagnoses for Birmingham, Cumbria, Northumberland and Durham suggest under-diagnosis in these regions. We show heightened genetic risk of haemochromatosis in people of Northwest Irish and Hebridean ancestry and suggest health-economic modelling of community screening should be targeted to these priority areas.

Hereditary haemochromatosis (HH) is an iron overload disorder that is one of the most common genetic conditions in people of European ancestry[1]. Homozygosity for a pathogenic variant in the *HFE* gene, resulting in the amino-acid substitution p.Cys282Tyr, is the major risk genotype for HH in European populations. The highest European frequencies of this allele are observed in the formerly Celtic-speaking populations in Ireland and in the United Kingdom[2,3], hence haemochromatosis is sometimes called the Celtic Curse. It occurs at particularly high rates in people of Irish ancestry[4,5] and analysis of ancient DNA[6] showed that an Early Bronze Age individual from Rathlin, an island off the coast of Northern Ireland, was a p.Cys282Tyr carrier. This means the allele existed at least 4000 years ago, and in a geographic region where it is found to be common today. It has been speculated

that this may relate to an evolutionary advantage when a low iron diet predominated[3] or during epidemics[7], although given such a diet and epidemics were widespread, it may be more likely that the high frequency is simply due to genetic drift over thousands of years. The symptoms of haemochromatosis range from chronic fatigue, joint and bone conditions to liver fibrosis, cirrhosis, hyperpigmentation and hepatocellular carcinoma. The risk of ill health and morbidity arising from haemochromatosis is higher in males than females, due at least in part to recurrent physiological blood loss in pre-menopausal women[8].

Early studies reported low penetrance of pathogenic *HFE* variants when two copies are present[9]. This has changed recently, as detailed phenotypic data from very large cohorts with long-term follow-up in electronic health records, such as the UK Biobank (UKB), have become

[1]Centre for Global Health Research, Usher Institute, University of Edinburgh, Edinburgh Bioquarter, UK. [2]MRC Human Genetics Unit, University of Edinburgh, Institute of Genetics and Cancer, Western General Hospital, Crewe Road, Edinburgh, UK. [3]Regeneron Genetics Center, Tarrytown, NY, USA. [4]School of Pharmacy and Biomolecular Sciences, Royal College of Surgeons in Ireland, Dublin, Ireland. [5]The SFI FutureNeuro Research Centre, Royal College of Surgeons in Ireland, Dublin, Ireland. [6]Centre for Genomic and Experimental Medicine, Institute of Genetics and Cancer, University of Edinburgh, Edinburgh, UK. ✉e-mail: jim.wilson@ed.ac.uk

available. The UKB is a cosmopolitan biomedical database containing genetic, lifestyle and health information from half a million UK participants[10]. Analysis of clinical outcomes to age 80[11] revealed that male and female p.Cys282Tyr homozygotes experienced greater excess morbidity than previously documented[12]. Particularly striking results in male homozygotes include a Hazard Ratio of 7.9 (5.5–11.4) for liver cancer and 2.6 (2.1–3.1) for any liver disease[11]. Much of this excess morbidity occurred after the age of 60 years; therefore, the life-course evidence on *HFE* penetrance has been expanded. Furthermore, analysis of outcomes is complicated by the fact that those participants with a diagnosis of haemochromatosis should have less excess morbidity if the condition were identified and treated early enough[13].

Such robust population cohort-based evidence (reviewed in ref. 14) prompted the addition in 2021 of homozygous p.Cys282Tyr variants in *HFE* to the ACMG v3.0 list of recommendations for reporting secondary findings in clinical exome and genome sequencing[15]. This was an important step in recognising the value of notification of this risk genotype when data is available. There is considerable benefit in identifying the overall genetic risk for HH, since the symptoms evolve over decades, and early diagnosis and treatment have been shown to mostly prevent the adverse consequences of iron overload[16]. Moreover, the opportunity to intervene and prevent disease is both simple and effective, primarily through regular venesection, which in turn can increase the supply of blood through donation.

The p.Cys282Tyr variant is the second most common pathogenic variant in *HFE*, while the most common is a variant that results in a different missense change in the protein, p.His63Asp. The main contribution to haemochromatosis risk of the p.His63Asp allele is through compound heterozygosity with p.Cys282Tyr, but with a penetrance about ten-fold lower than p.Cys282Tyr homozygotes[11]. A wide range of allele frequencies has been reported for p.Cys282Tyr in different populations worldwide, while p.His63Asp is more consistent in frequency, particularly across European populations. Approximately 1 in 156 people of European ancestry in the UK (as represented by the UK Biobank) are homozygous for p.Cys282Tyr[12], while around 1 in 40 are compound heterozygous for p.Cys282Tyr/p.His63Asp[11]. The frequency of these *HFE* risk genotypes is lower in southern Europeans and lower still in those with ancestry from outside Europe[17,18], but the extent of variation across the British Isles and Ireland has not been systematically assessed until now.

The International Classification of Diseases (ICD-10) code for haemochromatosis is E83.1 (Disorders of iron metabolism—haemochromatosis excluding anaemia). Lucas et al.[11] assessed the number of UKB participants of European ancestry who have this code, i.e., a diagnosis of haemochromatosis by the UK NHS, recorded in their electronic health record (EHR). Of the 451,270 UKB participants of European ancestry, at baseline (age range in the UKB at baseline was 40-69 years), 238 out of 267 men (89%) and 79 out of 87 women (91%) who had been diagnosed with haemochromatosis with code E83.1 carried one or two copies of the *HFE* pathogenic variants p.His63Asp and p.Cys282Tyr. When incident hospital diagnoses during a follow-up period of a mean of 13.3 years in the UKB participants were analysed[11], the additional numbers of people with both the code and at least one of the two most common pathogenic variants were 505 out of 590 men (86%) and 432 out of 476 women (91%). In total, 1254 out of the 1420 UKB participants (88%) with the E83.1 code had one or more copies of the p.Cys282Tyr or p.His63Asp alleles. These high percentages show that the code E83.1 provides a practical method of measuring hereditary haemochromatosis diagnoses in the NHS, but we note it is possible that in some cases the code only indicates the presence of the major risk genotype, rather than of iron overload.

It seems clear that targeting any population genetic screening to areas with greater numbers of p.Cys282Tyr homozygotes should improve cost-effectiveness. We therefore describe the detailed landscape of pathogenic *HFE* variation across the British Isles and Ireland,

along with the geography of haemochromatosis prevalence across England, to allow any work to build the evidence base for future screening programmes to prioritise areas of particularly high frequency, or discrepancies between overall genetic risk and prevalence.

## Results

### Pathogenic variants in HFE affecting risk of haemochromatosis in the UK Biobank

We undertook a detailed regional analysis of hereditary haemochromatosis risk across the British Isles and Ireland. The ClinVar database[19] contains a total of 27 variants in *HFE* that are recorded as pathogenic, short variants (< 50 bps), with a rating of at least 1* according to their system of classification (Supplementary Data 1). The Allele Frequency Browser (afb.ukbiobank.ac.uk) of whole genome sequence data from nearly 500,000 people in the UK Biobank allows assessment of allele frequency for any variant passing QC and with an ID reference, grouped by ancestry. 25 of the 27 pathogenic variants listed in ClinVar are either not detected or are ultra-rare in the UKB. Of those 25, the total number of alleles for the 13 variants with a minor allele count ≥1 is only 201, a combined allele frequency of $2.0 \times 10^{-4}$ (Supplementary Data 1). They will therefore only be seen vanishingly rarely as compound heterozygotes with the actionable variant p.Cys282Tyr (approximately 1/33,000 people in the UK will be compound heterozygotes of an ultra-rare pathogenic variant and p.Cys282Tyr). Interestingly, rare *HFE* variants as compound heterozygotes with p.Cys282Tyr were reported to be the most frequent cause of haemochromatosis in non-C282Y homozygous patients with the condition[20]. Nine private *HFE* variants, some of which are also found in UKB (Supplementary Data 1), were identified in 13 of 206 unrelated patients[20]. However, even when all are considered together, the collective allele frequency of these ultra-rare variants does not affect the landscape of pathogenic *HFE* allele variation in the British Isles and Ireland, beyond that shown by p.Cys282Tyr and p.His63Asp, and they were therefore not further analysed here.

The *HFE* variant c.845 G > A (chr6-26092913-G-A), which results in p.Cys282Tyr, has a total allele count of 71,949 out of 980,944 alleles in the UKB. Almost all (99%) of these were found in participants with genetic similarity to Non-Finnish Europeans, the ancestry group with by far the highest minor allele frequency of 0.078 (not shown). Researchers (e.g., ref. 11.) have therefore confined their studies of clinical outcomes of *HFE* risk genotypes in the UK Biobank to the European-heritage subset, and here we focus on the genomically British, in whom it is predicted that 1/164 will carry the major risk genotype (p.Cys282Tyr homozygote; Table 1).

The frequencies of p.Cys282Tyr are much lower in other population groups who live in the UK, particularly for non-European-heritage groups. Even for Continental Europeans, Italians are an order of magnitude less likely to carry the major risk genotype, whereas South and East Asian and African-heritage populations have predicted genotype frequencies three or more orders of magnitude lower. While still low risk, Afro-Caribbeans are noticeably more at risk (1/16,000) than other sub-Saharan African-heritage populations (< 1/1 M). In contrast, the more common pathogenic variant *HFE* c.187 C > G (chr6-26090951-C-G), which results in p.His63Asp, varies little across European-heritage populations, but is also much rarer particularly in African- and East Asian-heritage populations (Table 1).

### Non-HFE haemochromatosis variants

Variants in five further genes have been reported to cause other forms of hereditary haemochromatosis: *SLC4OA1* [MIM #604653], *TFR2* [604720], *HAMP* [606464], *HJV* [608374] and *FTH1* [134770]. In total, 1256 minor alleles are observed (including only one homozygote), for a combined allele frequency of only 0.0014 (Supplementary Data 2). Variants in all genes but *SLC4OA1* (for which there are 96 minor alleles observed) act recessively. Thus, similarly to the ultra-rare *HFE* variants,

**Table 1 | Frequencies of the *HFE* c.845 G > A (C282Y) and c.187 C>G (H63D) variants in the UKB by population group**

| Group | *N* | c.845 G>A (p.Cys282Tyr) | | | | c.187 C > G (p.His63Asp) | |
|---|---|---|---|---|---|---|---|
| | | MAC | MAF | q2 | one in | MAC | MAF |
| Genomically British | 408,780 | 63,763 | 0.0780 | 0.00608 | 164 | 123,466 | 0.1510 |
| German | 2088 | 254 | 0.0608 | 0.00370 | 270 | 617 | 0.1477 |
| French | 801 | 80 | 0.0499 | 0.00249 | 401 | 285 | 0.1779 |
| Polish | 612 | 42 | 0.0343 | 0.00118 | 849 | 182 | 0.1487 |
| Italian | 808 | 31 | 0.0192 | 0.00037 | 2.7k | 230 | 0.1423 |
| Ashkenazi Jewish | 2868 | 71 | 0.0124 | 0.00015 | 6.5k | 724 | 0.1262 |
| Afro-Caribbean | 2171 | 34 | 0.0078 | 0.00006 | 16k | 70 | 0.0161 |
| Sri Lankan | 654 | 3 | 0.0023 | 0.00001 | 190k | 133 | 0.1017 |
| British Indian | 5317 | 23 | 0.0022 | 0.00000 | 213k | 834 | 0.0784 |
| British Pakistani | 1645 | 4 | 0.0012 | 0.00000 | 676k | 239 | 0.0726 |
| Kenyan | 1049 | 2 | 0.0010 | 0.00000 | 1 M | 170 | 0.0810 |
| East Asian | 2244 | 2 | 0.0004 | 0.00000 | 5 M | 134 | 0.0299 |
| West African | 1317 | 0 | 0.0000 | 0.00000 | >6 M | 15 | 0.0057 |

The groups in Table 1 are ordered by decreasing MAF of p.Cys282Tyr. Genomically British were identified by UK Biobank (Data field 22006). Otherwise, data were taken from UKB volunteers either reporting birthplaces in Germany, France, Poland, Italy, Sri Lanka or Kenya, who matched the majority ancestry group from that nation using dbscan (Methods), or those self-declaring their ethnicity as Afro-Caribbean, British Indian, British Pakistani or West African. Ashkenazi Jewish and East Asian frequencies are taken from the UKB allele frequency browser, because this ancestry was not available through self-report in UKB and there were too few individuals born in any single East Asian nation, respectively. In this way, the major groups living in the UK (with >500 volunteers in the UKB) are represented.
*MAC* minor allele count, *MAF* minor allele frequency, *q2* predicted frequency of p.Cys282Tyr homozygotes, themajor risk genotype; 'one in' gives the ratio of people in each population group who carry the major risk genotype, k thousand, M million.

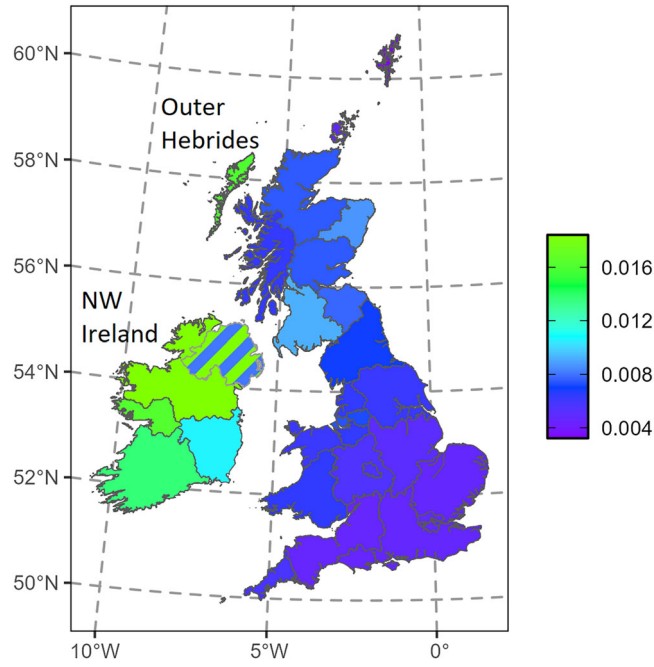

**Fig. 1 | Map of expected homozygote frequencies for the major risk variant, *HFE* p.Cys282Tyr, across Britain and Ireland.** The square of the minor allele frequency is plotted for 29 regions. The two communities in Northern Ireland are represented by hatching.

the non-*HFE* variants will not have a significant effect on the geography of hereditary haemochromatosis risk across the British Isles and Ireland.

### Population frequencies of the HFE p.Cys282Tyr and p.His63Asp variants in regions of the British Isles and Ireland

We divided the British Isles and Ireland into 29 geographic regions, 24 in Britain and its associated islands, and 5 in Ireland (the latter derive from the 7574 people in UKB who report they were born in the Republic of Ireland or Northern Ireland and that their ethnicity is Irish).

The British samples were assigned to a region according to their birthplace (13 in England, 2 in Wales and 9 in Scotland). In the absence of birthplace information for the Irish samples, they were divided into five geographic groups on the basis of genetic clustering with Irish samples of known grandparental ancestry ("Methods").

We focussed first on predicted p.Cys282Tyr homozygote genotype frequencies, which are more accurate than depending only on the low counts of observed homozygotes, as it takes the >20-fold more numerous heterozygotes into account (both variants are in Hardy-Weinberg equilibrium in the white British sample used here, $p > 10^{-4}$, $\chi^2 = 11.3$). The predicted genotype frequency and therefore proportion of people at risk varies by nearly 6-fold across the regions of Britain and Ireland (Fig. 1 and Supplementary Data 3).

In England, there is a decreasing cline of predicted p.Cys282Tyr homozygotes from north to south, with North, Northwest, Merseyside and Manchester all being higher than the rest of England (overall 1/142), with London, Southwest, Southeast and East England having the lowest frequencies (overall 1/212, significantly lower than the northern regions, $p < 0.00001$, $\chi^2 = 360$). Yorkshire, the Midlands and Birmingham have intermediate frequencies (overall 1/175). There is a weaker west to east cline, from Wales (1/156) through the Midlands to East Anglia ($p < 0.00001$, $\chi^2 = 26.0$ for Wales-East Anglia difference).

Mainland Scotland's frequencies of predicted p.Cys282Tyr homozygotes increase beyond north English levels (overall 1/117), with a maximum in Southwest Scotland (including Glasgow), which is also the highest in Mainland Britain (1/108, significantly higher than northern England, $p < 0.00001$, $\chi^2 = 58.0$). Frequencies in Ireland are higher still (overall 1/75), however there is considerable variation across communities. An ancestry group found in Northern Ireland and also Lowland Scotland has the lowest frequency (1/123) while the highest is in the Northwest Irish group (1/54, significantly higher than SW Scotland, $p < 0.00001$, $\chi^2 = 55.0$), found across Northern Ireland, including also a broad sweep of the northern counties of the Republic of Ireland, from north Mayo to Meath, up to Donegal. The Outer Hebrides have the highest frequency in the UK (1/62, significantly higher than SW England, $p < 0.00001$, $\chi^2 = 58.8$), and higher than Northern Ireland (overall 1/71). The Northern Isles of Scotland have low frequencies, Orkney (1/209) being similar to southern England, while Shetland (1/309) is the lowest of all the British Isles and Ireland.

**Table 2 | Prevalence of Haemochromatosis Diagnosis by Ethnicity in NHS England**

| Ethnicity | E83.1 | Total | Prevalence (%) | 1 in |
|---|---|---|---|---|
| White Irish | 1355 | 225,320 | 0.6014 | 166 |
| White British | 53,575 | 32,380,055 | 0.1655 | 604 |
| Chinese | 310 | 241,320 | 0.1285 | 778 |
| White—Any other White background | 2470 | 3,279,680 | 0.0753 | 1328 |
| Black or Black British—Caribbean | 325 | 437,095 | 0.0744 | 1345 |
| Black or Black British—African | 675 | 1,047,810 | 0.0644 | 1552 |
| Any Other Ethnic Group | 925 | 1,440,830 | 0.0642 | 1558 |
| Black or Black British—Any other Black background | 255 | 399,380 | 0.0638 | 1566 |
| Asian or Asian British—Pakistani | 590 | 1,270,035 | 0.0465 | 2153 |
| Asian or Asian British—Any other Asian background | 430 | 933,260 | 0.0461 | 2170 |
| Asian or Asian British—Bangladeshi | 180 | 450,140 | 0.0400 | 2501 |
| Asian or Asian British—Indian | 490 | 1,319,040 | 0.0371 | 2692 |
| Mixed—All Mixed backgrounds | 365 | 1,070,945 | 0.0341 | 2934 |
| All (including ethnicity unknown and not stated) | 70,365 | 63,664,825 | 0.1105 | 905 |

Data are shown for all ages, genders and regions, sorted by raw prevalence.

The p.His63Asp allele varies only 1.4-fold across Britain and Ireland, from a maximum in north and northeast Scotland (17.5%) to a minimum in the Outer Hebrides (12.7%). All other Mainland Scottish, Irish, Welsh and English regions have frequencies between 13.4%-15.9%. The majority of the risk of haemochromatosis arising from the p.His63Asp variant is conferred through compound heterozygosity with p.Cys282Tyr, hence we have mapped out the variation in the predicted genotype frequency of the compound heterozygote (Supplementary Fig. 1). The more even distribution of p.His63Asp means there is less geographic variation predicted for the compound heterozygote, and it is correlated with that for the p.Cys282Tyr homozygote to give a very similar landscape: the highest frequency is also in the Northwest Irish group and the lowest in Shetland.

We can use the cumulative incidence for haemochromatosis to age 80 years for each genotype (calculated in UK Biobank[11], that is 0.56 for p.Cys282Tyr homozygotes and 0.06 for p.Cys282Tyr/p.His63Asp compound heterozygotes (in European-heritage males), to calculate a combined genetic risk of haemochromatosis in each region (Supplementary Fig. 2). This is very strongly correlated with the predicted p.Cys282Tyr homozygote frequency ($r = 0.999$), and thus shows the same clines across England and Wales, and heightened risk in Scotland and Ireland. The highest risk in the UK is found in the Outer Hebrides, where 1/94 males are predicted to develop iron overload by the age of 80 years, while in the Northwest Irish group, 1/80 males are predicted to develop the condition. In Northern Ireland overall, 1/133 men are predicted to develop iron overload, with 1/103 in the Republic of Ireland, 1/161 in Mainland Scotland, 1/208 in Wales and 1/226 in England.

### Prevalence of a diagnosis of haemochromatosis in England using NHS DigiTrials

The total number of people in the UK with a current diagnosis of haemochromatosis has been difficult for researchers to ascertain. Symptoms are dependent on age and sex, and there is also considerable evidence from the UKB[11,12] and elsewhere[21] that many individuals with homozygous p.Cys282Tyr genotypes who are clinically affected have not been identified and diagnosed. The charity Haemochromatosis UK reported that NHS acute trusts and health boards were aware of a total of 20,698 people in the UK who were receiving care for genetic haemochromatosis in 2018/19 (https://www.haemochromatosis.org.uk/state-of-the-nation-2020). However, this figure is subject to considerable potential error, as 37% of NHS Trusts were unable to provide figures of people diagnosed, and some of the

data provided relate to patient episodes, rather than individual patients. Accurate knowledge of the number of people with hereditary haemochromatosis in each area of the UK is clearly important for commissioning effective clinical care and, in conjunction with genetic data, may help identify regions particularly affected by under-diagnosis.

We therefore made use of the recently available NHS Digitrials resource ("Methods"). Symptoms due to type 1 haemochromatosis rarely develop before 30 years of age, instead usually becoming apparent between 40-60 years[11]. Consistent with that, only 3.1% of the White British individuals with haemochromatosis in the Digitrials dataset were aged 30 years or younger at their most recent diagnosis. We note that the data in DigiTrials provides only the age group range at the time of the most recent recording of the condition. Counts of E83.1 case numbers as a population percentage in the NHS DigiTrials White British dataset were analysed for males and females separately. As expected from the epidemiology of the condition, we determined the prevalence to be higher in males (59.2% of diagnoses) than in females.

We found that on 31/10/2024, a total of 70,365 people of all ages, ethnicities and genders had the EHR code E83.1 for haemochromatosis across all 42 NHS England Integrated Care Boards (Table 2). These are among the 63,664,825 people in the DigiTrials dataset (Table 2). There is little variation in prevalence across the ten deciles of deprivation—among white British, the lowest prevalence decile (second most deprived) has 0.154% and the highest prevalence decile (third least deprived) 0.173%, a factor of only 12% (Supplementary Data 4).

Among the self-declared ethnicities, the highest prevalence of haemochromatosis is among the White Irish (Table 2), of whom there are 225,320 in the dataset. The prevalence of 1 in 166 is 3.7-fold higher than in the second-highest ethnic group, which is White British. In contrast, the prevalence for a different ICD-10 code, G35 (multiple sclerosis, $n = 88,015$ in White British) is virtually identical between the two groups (data not shown). Similarly, little variation in prevalence between the White British and White Irish groups in the DigiTrials dataset was reported for seven common complex conditions[22]. The prevalence of E83.1 in other ethnic groups are much lower, ranging from 1/778 for the Chinese to 1/2934 for all Mixed backgrounds (Table 2).

We further analysed the 53,575 White British individuals with a diagnosis of haemochromatosis by Integrated Care Board (ICB). The prevalence by population shows a variation of nearly 4-fold across these 42 areas of England, with the lowest in Gloucestershire, Somerset, Norfolk & Waveney and Kent & Medway, in which 1/1001–1/1137

**Table 3 | Regional prevalence of haemochromatosis among white British in England**

| Region | E83.1 | Total | Prevalence (%) | 1 in |
|---|---|---|---|---|
| Cheshire & Merseyside | 6635 | 1,924,680 | 0.345 | 290 |
| Manchester | 3700 | 1,730,190 | 0.214 | 468 |
| London | 4270 | 2,184,795 | 0.195 | 512 |
| Lancashire & S Cumbria | 2190 | 1,167,890 | 0.188 | 533 |
| Yorkshire | 5970 | 3,569,205 | 0.167 | 598 |
| W-Midlands | 3665 | 2,259,905 | 0.162 | 617 |
| North | 3645 | 2,277,515 | 0.160 | 625 |
| Cornwall | 760 | 478,210 | 0.159 | 629 |
| E-Midlands | 4715 | 3,009,140 | 0.157 | 638 |
| East | 5300 | 3,993,515 | 0.133 | 753 |
| Southwest | 4530 | 3,459,385 | 0.131 | 764 |
| Southeast | 6720 | 5,176,910 | 0.130 | 770 |
| Birmingham | 1455 | 1,144,980 | 0.127 | 787 |

Data are shown for all ages and genders, sorted by raw prevalence.

**Table 4 | Regional comparison of combined genetic risk and prevalence of haemochromatosis in white British across England**

| Region | Prevalence (%) | Genetic risk (%) | Ratio |
|---|---|---|---|
| Cheshire & Merseyside | 0.345 | 0.542 | 1.57 |
| London | 0.195 | 0.376 | 1.93 |
| E-Midlands | 0.157 | 0.388 | 2.48 |
| Manchester | 0.214 | 0.532 | 2.49 |
| W-Midlands | 0.162 | 0.421 | 2.60 |
| Southwest | 0.131 | 0.356 | 2.72 |
| East | 0.133 | 0.364 | 2.74 |
| Cornwall | 0.159 | 0.441 | 2.78 |
| Yorkshire | 0.167 | 0.467 | 2.79 |
| Lancashire & S Cumbria | 0.188 | 0.525 | 2.80 |
| Southeast | 0.130 | 0.366 | 2.82 |
| North | 0.160 | 0.519 | 3.24 |
| Birmingham | 0.127 | 0.427 | 3.36 |

Prevalence is the raw prevalence in each region; genetic risk is calculated as the product of the penetrances and genotype frequencies, summed for p.Cys282Tyr homozygotes and p.Cys282Tyr/p.His63Asp heterozygotes; both are given as %. Penetrances for males were used for this calculation.

people are diagnosed, and the highest in Cheshire & Merseyside, where 1/290 people are diagnosed (Supplementary Data 5). There is a considerable drop in prevalence in the next three ICBs, North Central London, North West London and Greater Manchester, all with 1/454-1/468 being diagnosed.

Data are also available at the level of 134 English NHS Trusts, which clearly show the peak of diagnoses in Merseyside: among White British, the top seven trusts are all in Liverpool/Merseyside, varying from 1/177 to 1/278 prevalence (Supplementary Data 6). At the opposite end, trusts in Kent, Norfolk and Gloucestershire all have <1/1100 diagnoses, with Medway NHS Foundation Trust having only 1/1972 diagnoses of E83.1, 11-fold fewer than the Liverpool Heart and Chest Hospital NHS Foundation Trust. To make comparisons more straightforward, we have further grouped the data into the same 13 larger regions of England as used for genetic analysis (Table 3). Manchester and London follow Merseyside at the high prevalence end of the table, while Birmingham is at the low end, along with the southeast and southwest of England.

## Discrepancies between the combined genetic risk and prevalence

Unfortunately, haemochromatosis prevalence data from the NHS are not available for Scotland, Wales and Northern Ireland. Within England, we find a general agreement between the predicted frequency of those at genetic risk (among genomically British in UK Biobank) and the raw prevalence of diagnoses among white British in DigiTrials ($r = 0.65$ between raw prevalence and combined genetic risk from p.Cys282Tyr and p.His63Asp; Table 4). However, comparison of the ratio of combined genetic risk to raw prevalence reveals Birmingham and North England (North Cumbria, Northumberland, Durham) to have the highest ratios (3.36 and 3.24, respectively), that is, a much lower prevalence than would be expected from their combined genetic risks, suggesting under-diagnosis in these areas. At the other end of the scale, Cheshire & Merseyside and London have the lowest ratios (1.57 and 1.93, respectively), suggesting diagnostic rates in these regions are up to twice as good as in the worst regions.

## Discussion

Inclusion of *HFE* p.Cys282Tyr homozygotes in the ACMG secondary findings v3.0 list, following more reliable life-course estimates of penetrance, raised awareness of the genetic risk of haemochromatosis, particularly with respect to older males who are most likely to develop complications. Return of actionable genotype results does not entail significant additional costs, if the data have been generated for other purposes (except for downstream costs for identified individuals), and should improve health outcomes for the p.Cys282Tyr homozygotes. However, few population research cohorts, including the UK Biobank, currently return such results to their participants. An exception is the Viking Genes cohort, in which we have returned results to a total of 49 p.Cys282Tyr homozygotes living across the UK, using approved processes[23]. Outcomes following receipt of their result by the research participants are being collected and will be reported in due course.

Our comprehensive analysis of pathogenic haemochromatosis variants across the British Isles and Ireland reveals that, despite the existence of numerous ultra-rare variants, geographic variation in risk is driven almost entirely by the landscape of the major p.Cys282Tyr variant. The p.Cys282Tyr variant varies by 6-fold across the UK and Ireland. The highest frequency in mainland Britain is in SW Scotland, around Glasgow. The lowest frequency in the British Isles is found in the Northern Isles of Scotland, and it is tempting to speculate that the lower frequencies observed there and in England may be caused by dilution of an earlier gene pool, high in p.Cys282Tyr, by incoming Norse Vikings in the Northern Isles[24] and Anglo-Saxons or others in England[25,26].

We recognise that no volunteer cohort can be a perfect representation of the population from which it is drawn. There is evidence of a healthy volunteer selection bias in the UKB[27], but we consider it unlikely that these biases operated to different degrees in different parts of the UK, such that they would substantively affect our conclusions.

Analysis of grandparental origins within the Outer Hebrides shows that the p.Cys282Tyr variant is found at frequencies over 12% all the way from Barra and Uist in the south to Lewis in the north (not shown). We note that the estimated frequency is also very high in the Isle of Skye (15%), using the small sample available (n = 46 with four grandparents from Skye). Further study is thus required to assess the degree to which Sgiathanachs (people from the Isle of Skye) would also potentially benefit from screening.

While the frequencies are high everywhere in Ireland, they are highest for an ancestry group found in the Northwest of Ireland, including the 6 counties of Northern Ireland and at least part of 11 counties in the Republic of Ireland, in which 1/54 people harbour the homozygous risk genotype. We note that another ancestry group found in Northern Ireland (but also present in Lowland Scotland[24]), has

the lowest frequency in Ireland (1/123; significantly different from NW Irish, $p < 0.00001$, one-sided $\chi^2$). These two genetic clusters almost certainly reflect the historic Ulster Scots (Protestant) and indigenous (Roman Catholic) communities in Northern Ireland. The ancestors of the Ulster Scots arrived in Ulster starting over 400 years ago in the Plantation of Ulster, but continuing in subsequent decades, the majority of whom came from Lowland Scotland (and some of whom are known as Scotch-Irish in the USA), with others from Northern England. In the intervening centuries, the communities have remained sufficiently structured to allow identification of the two gene pools and their different allele frequencies today. Indigenous NW Irish have the highest overall genetic risk of haemochromatosis in the world (1/80 predicted to develop the disease by age 80 years), while Ulster Scots have half that risk (1/172), similar to that across much of Mainland Scotland. We can speculate that differences may also be found between indigenous and Irish-origin communities in West Central Scotland, for example, in Larkhall and Coatbridge.

The NHS Digitrials database is a powerful, unbiased resource for analysis of the prevalence of coded conditions and for uncovering areas of potential under-diagnosis. In England, the highest prevalence is seen among self-reported white Irish, where the disease is almost four times more common than in the white British, who in turn have twice as much haemochromatosis as any other ethnic group apart from the Chinese. There is also considerable variability within the white British, from a low in Kent in SE England to a high in Merseyside/Liverpool in the NW of England.

Regions with a disconnect between the combined genetic risk and the population rate of haemochromatosis diagnosis, such as Birmingham, North Cumbria, Northumberland and Durham are potential targets for awareness-raising about the condition, both among the public and health-care workers. Our analysis assumes even penetrance across different regions, but it remains unclear if differences in dietary iron intake could also influence the results[28]

While comparison of genetic risk and prevalence indicates high diagnostic rates in Merseyside, we can also hypothesise that the high prevalence and high genetic risk there may originate in part from the historically large Irish diaspora in and around Liverpool. The Merseyside & Cheshire Integrated Care Board reports 0.32% white Irish, just below the average for England (0.35%), and this varies widely in Merseyside Trusts from 0.21% in Warrington and Halton to 0.62% in Liverpool Women's NHS Foundation Trust. However, >20% of Liverpool's population was Irish in the 1851 census (https://web.archive.org/web/20081007071211/http://www.liverpoolmuseums.org.uk/nof/emigrants/access/liverpool.asp?%5Blookup%5D=irish), the descendants of whom, five generations later, would be classed as white British. Further support comes from the localised increase in the frequency of the mostly Irish Y chromosome marker R1b-M222 in Liverpool[29] (https://doi.org/10.7488/ds/3472), at 5.2% among genomically British in Liverpool versus 2.5% across England generally. The assimilation of large numbers of Irish immigrants may therefore explain Merseyside having the highest frequency of p.Cys282Tyr in England. Similar processes may also explain why Glasgow and Southwest Scotland have the highest frequencies in Mainland Britain, given the longstanding immigration from Ireland.

The DigiTrials Unknown ethnicity category (Table 2) contains two distinct groups, ~4 M with no ethnicity recorded, and ~7 M registered with a GP but with no hospital episode statistics. We found that for all ethnicities and for multiple diseases queried, the prevalences were always lowest among the Unknown ethnicity category (not shown). This may be due to issues with record linkage among a subset, or over-representation of very healthy individuals who have not been a hospital in- or out-patient since 1989. If the latter were the case, then our prevalence estimates for the declared ethnicities would be slightly biased upwards, due to these people not being counted. However, the overall figures for the entire 63 M dataset would not be inflated.

It is interesting to note the higher-than-expected prevalence of haemochromatosis diagnoses in the Chinese in England (1/778; Table 2), given the vanishingly low frequency of the *HFE* p.Cys282Tyr variant (predicted 1/5 M homozygotes among East Asians in UKB). Our survey of the non-*HFE* genes revealed the pathogenic *SLC40A1* p.Ser209Leu variant to be 35 times more common among East Asians (1/187 are carriers) than among Europeans in UKB (Supplementary Data 2). As *SLC40A1* variants show dominant inheritance for haemochromatosis type 4 (MIM #606069), it suggests this variant may be an important contributor to risk in East Asians. However, the penetrance is not well understood.

The inclusion of *HFE* p.Cys282Tyr in the ACMG secondary findings list, together with the substantial increases in risk of liver disease detected in the UKB analyses, further motivated investigation of the utility of genetic screening[21]. In the Geisinger MyCode study of 86,601 participants from Pennsylvania, USA, 72% of p.Cys282Tyr homozygotes learned of their genotype through screening, and of these 37% had iron overload[21], confirming the ability of genetic screening to detect iron overload and allow relevant pathways of care to be accessed. Studies of the cost-effectiveness of haemochromatosis screening have been published and reviewed[30]. Population screening for *HFE* p.Cys282Tyr homozygosity is not entirely straightforward to implement, since it requires societal acceptability, uptake by health professionals, access to testing, and access to appropriate management. Nonetheless, the increasingly clear evidence for excess morbidity and mortality of p.Cys282Tyr homozygotes is leading to calls for genotyping as a preventative tool for earlier diagnosis of haemochromatosis[11,21,31,32]. A recent opinion piece makes a convincing case for population screening for hereditary haemochromatosis, specifically p.Cys282Tyr homozygotes[33]. While we advocate that screening should be available to all, given the resource implications, the data presented here suggest prioritisation of consideration of communities for screening, for example, through health-economic analyses, in a variety of ways. The very high combined genetic risk apparent for Northern Irish, Outer Hebrideans, across the Republic of Ireland and SW Scotland around Glasgow suggests a focus on these areas would discover the maximum number of at-risk individuals. Within England, the mismatch between the observed prevalence and that expected from the underlying genetics can be used to highlight further regions where screening could reveal additional cases, such as Birmingham and regions of England to the north of Yorkshire.

There are extensive Irish diasporas in the New World, for example, nearly 5 M Irish immigrated to the USA. The increased frequencies of the p.Cys282Tyr variant and hence risk of haemochromatosis will have been carried to the New World with them, suggesting that people with significant Irish ancestry ought to be aware of a potentially heightened risk of the disease. The same is true of Hebridean descendants, for instance, in Nova Scotia, Canada.

Using data from over 400,000 individuals, we provide evidence for significant differences in the risk of hereditary haemochromatosis across the British Isles and Ireland, with global maxima among the northwest Irish and Outer Hebrideans. Local frequency peaks are observed in Glasgow/SW Scotland and Liverpool, both likely in part due to Irish ancestry, and corroborating the Celtic Curse moniker. In England, the prevalence of haemochromatosis follows the pattern of genetic risk, but outliers in Birmingham and the northernmost counties of England suggest under-diagnosis in these regions. The map of haemochromatosis risk will allow work underpinning future genetic screening programmes to be accurately targeted to the most at-risk populations first, in order to maximise the benefits by averting liver cancers, cirrhosis, arthropathy and other negative outcomes of the disease.

## Methods

### Study populations

*HFE* allele frequencies were measured in two population cohorts[10,34] using a total of ~440,000 research volunteers with ancestry covering

all regions of the British Isles and Ireland, together with all substantial minorities within the UK.

The UK Biobank contains genetic, lifestyle and health information from half a million UK participants[10], recruited in urban centres. Viking Genes comprises three island-ancestry cohort studies—the Orkney Complex Disease Study (ORCADES), VIKING I and VIKING II/III. ORCADES contains more than 2000 deeply phenotyped and exome-sequenced research subjects with three or four self-reported grandparents from Orkney[35], whereas VIKING I is a similar-sized cohort of participants from Shetland[36]. VIKING II/III is a worldwide cohort of 6000 people, recruited online, with ancestry (two or more grandparents) from the Northern or the Western Isles of Scotland[34].

## Genotyping and DNA sequencing

Details of genotyping, sample and variant quality control of UK Biobank genotyping data are described in Bycroft et al.[37]. The exome sequence dataset of 2113 VIKING I[38,39] and 5106 VIKING II/III sequences passed all genotype quality control thresholds (including removal of samples with disagreement between genetically determined and reported sex, high rates of heterozygosity or low sequence coverage, and removal of sites read depth <7 or allele balance <15%), as described for the 2097 exomes in the ORCADES cohort[40]. The 5106 VIKING II/III sequences are reported for the first time in this manuscript. All Viking Genes sequence data were prepared at the Regeneron Genetics Center, following the process detailed for UK Biobank[41]. Imputed genotypes were used for UK Biobank (data field F.22828); p.Cys282Tyr and p.His63Asp had impute scores of 0.997 and 1.00, respectively.

Pathogenic *HFE* genotypes for all 48 individuals from the Viking II/III exome dataset were verified by a different sequence analysis method. The *HFE* c.845 G > A (chr6-26092913-G-A) variant was confirmed in Viking II/III p.Cys282Tyr homozygotes by a Taqman assay (Applied Biosystems by Life Technologies predesigned assay ID C_1085595_10, results analysed using QuantStudio Design & Analysis Software). Twelve further p.Cys282Tyr homozygotes from ORCADES and VIKING I were validated using Sanger sequencing[23].

## Identification of ethnic groups and minority populations in UK Biobank

Data analysis used R version 4.4.2. We quantified *HFE* variant frequencies in the 13 largest non-white-British or -Irish population groups in the UK by first selecting individuals from the UKB matching country of birth (UKB data field F.20115) or ethnic background (field F.21000). For each group, we then selected the core genetic cluster of individuals from that group using the dbscan algorithm implemented in the R package dbscan (version 1.2–0). We applied this to the co-ordinates from the top two principal components (field F.22009), selecting the DBSCAN cluster consistent with that population group. We did not assess any group with under 500 volunteers. The Irish-born individuals in the UK Biobank who identified as Irish were assigned to five genetic clusters on the basis of identity-by-descent (IBD) segment sharing. UK Biobank genotypes (UK Biobank data field F.22418) were first filtered using PLINK v2[42] for variants and individuals with missingness <5%, variants with a minor allele frequency >1% and a Hardy-Weinberg Equilibrium $p$-value $> 1 \times 10^{-9}$. The last used the PLINK flag --hwe with the additional flags midp and keep-fewhet. Individuals with a heterozygosity >3 standard deviations from the mean were subsequently removed. Relatives closer than 3rd degree were identified with KING[43] version 2.3.2, and one from each pair was removed. This cleaned genotype dataset was subsequently phased with Eagle version 2.4.1[44] using human genome build GRCh37, and IBD segments with a length >3 cM were detected with hap-ibd[45] version June 14, 2023. Using IBD segments >3 cM and <30 cM, we calculated the total sum length of IBD segments between all individuals and performed Leiden network community detection[46] using the python implementation in the leidenalg package (version 0.10.2). We annotated the five detected clusters using samples with known grandparental birthplace, from the Irish DNA Atlas[47], People of the British Isles[25] and the Trinity Student dataset[48] based on Irish Provincial identity of these references and shared principal component analysis (PCA) co-ordinates. These PCA co-ordinates were estimated in PLINK v2 from a merged genotypes dataset of the UKB Irish and Irish-British references following the same quality control thresholds as above. PCA was carried out using a pruned set of variants filtered for linkage disequilibrium identified with the PLINK --indep-pairwise 1000 50 0.2 command. PCA was subsequently performed using the PLINK --pca command. All first and second degree relatives in Viking Genes were removed using PLINK v1.904b and PRIMUS v1.9.0.

## Non-HFE haemochromatosis variants

Using ClinVar, we created a catalogue of 174 pathogenic or likely pathogenic variants that were associated with some form of haemochromatosis with at least 1* evidence (no such variants were observed in *FTH1*). We assessed the frequencies of 56 of these variants present in the UK Biobank using the allele frequency browser.

## NHS DigiTrials

The NHS DigiTrials Feasibility Self-Service allows researchers to search routinely collected healthcare records from over 63 million patients across NHS England national datasets in a secure environment, to count and locate relevant individuals without identifying them (https://digital.nhs.uk/services/nhs-digitrials/feasibility-service). Data are searchable for people who are alive, have a valid NHS number, are registered with a GP in England, and whose demographic record is not flagged as sensitive or invalid. Individuals are recorded once without duplicates, and only the latest episode is available from the database. Hospital Episode Statistics (HES) data were available for people diagnosed between 01/04/1989 and 31/10/2024. The NHS DigiTrials data can be divided geographically, and can also be queried by gender, ethnicity, age and socio-economic status. Ethnicity is self-reported using the 2001 UK census classification. Data from queries was downloaded through the Safe Output Service, rounded to the nearest increments of 5. We note that the genetic data can be studied according to birthplace, whereas the DigiTrials regional data is recorded by place of residence.

## Data handling and statistical testing

We visualised the variant frequencies across Great Britain and Ireland using shapefile data implemented in R, processed using the package sf, and graphically visualised using the ggplot2 package version 3.5.1. We sourced Irish county boundary data from OpenStreetMap (https://openstreetmap.org) and British International Territorial Level 3 Boundaries from the UK Office for National Statistics (https://www.data.gov.uk/dataset/9f55fc2c-f90e-49b3-9c17-f96d8240b4f1/international-territorial-level-3-january-2021-boundaries-uk-bfc-v31). Differences in genotype frequencies and Hardy-Weinberg equilibrium were tested for significance using one-sided $\chi^2$ tests with one degree of freedom.

## Ethical approval

Eligible participants were recruited to ORCADES, the Viking Health Study Shetland (VIKING I), and to VIKING II/III, collectively Viking Genes[34], South East Scotland Research Ethics Committee, reference 19/SS/0104. This research has been conducted using data from the UK Biobank (approved by North West Centre for Research Ethics Committee, reference 11/NW/0382), as part of project numbers 103770 and 19655. All cohort participants gave written informed consent for research procedures that included DNA sequencing. Some frequencies for *HFE* variants reported in Table 1, Supplementary Data 1 and Supplementary Data 2 were derived from the UK Biobank Whole Genome Sequencing (WGS) project and were obtained from the UK Biobank

Allele Frequency Browser (afb.ukbiobank.ac.uk), which was generated by the WGS consortium under the UK Biobank Resource (project ID 52293).

## Reporting summary

Further information on research design is available in the Nature Portfolio Reporting Summary linked to this article.

## Data availability

There is neither Research Ethics Committee approval nor consent from participants to permit open release of the individual-level research data underlying this study. Instead, the research data are available through managed access after application to each population cohort. For UK Biobank, www.ukbiobank.ac.uk/register-apply; for Viking Genes, https://viking.ed.ac.uk/our-data-and-samples/access; in accordance with the consent given by participants and existing favourable opinions from the relevant Research Ethics Committee. Requests will be responded to within 2 weeks. Each Data Access Committee (DAC)-approved project is subject to a data or materials transfer agreement (D/MTA) or commercial contract. Data may then be shared with academic or commercial recipients worldwide and may be used within the parameters of the study Protocols[34,37]. The 5106 VIKING II/III exome sequences were generated as part of the study. Data from routinely collected healthcare records on patients across NHS England national datasets is available on application to the NHS DigiTrials Feasibility Self-Service (https://digital.nhs.uk/services/nhs-digitrials/feasibility-service). Access to the UK Biobank genotype data was approved under applications 19655 and 103770. All other data supporting the findings of this study are available in the article and its Supplementary Information files.

## Code availability

All bespoke code has been deposited in GitHub (https://github.com/viking-genes/uk-hfe). All other software is detailed in the Methods.

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

## Acknowledgements

The authors wish to thank all the volunteers who participated in the research cohorts underlying this study. Participants are extensively involved and engaged in both the UK Biobank and Viking Genes. This work used data provided by patients and collected by the NHS as part of their care and support, reused through the DigiTrials resource with the permission of NHS England. Map data is copyrighted by OpenStreetMap contributors and is available from https://openstreetmap.org under the Open Database License. The Viking Genes Taqman genotyping was performed at the Edinburgh Clinical Research Facility. We thank Gianpiero L. Cavalleri for access to the Irish DNA Atlas and Trinity Student Dataset and Sir Walter Bodmer for access to the People of the British Isles, for use in defining genetic clusters among the UKB Irish. We thank Ashwini Shanmugam, Lucija Klaric and Caroline Hayward for assistance with data and helpful discussions. This work was funded by the Fernau Medical award 2024 from the charity Haemochromatosis-UK (to JFW), an MRC University Unit award to the MRC Human Genetics Unit, University of Edinburgh, MC_UU_00007/10 (to JFW), and a Wellcome Trust Institutional Translational Partnership Award (University of Edinburgh 222060/Z/20/Z -PIII031; to JFW).

## Author contributions

S.M.K. managed the project, did the NHS DigiTrials analysis and drafted the manuscript. B.S.F. analysed Viking Genes genotypes and performed look-ups in UKB. G.T. and A.R.S. conceived and managed the Viking Genes exome sequencing. E.G. performed a range of geographic and ancestry analyses and interpreted the results. J.F.W. was awarded funding to implement the work, conceived the hypotheses, interpreted the data and co-wrote the manuscript. All authors provided input and feedback on drafts of the manuscript.

## Competing interests

A.R.S. and G.T. are employees and stock and stock option holders of Regeneron Pharmaceuticals, Inc. The remaining authors declare no competing interests.
