## [Transparent Peer Review file · Nature Communications]

The landscape of hereditary haemochromatosis risk and diagnosis across the British Isles and Ireland

Corresponding Author: Professor James Wilson

Version 0:

Reviewer comments:

Reviewer #1

(Remarks to the Author)

Kerr and colleagues present data on the prevalence of HFE p.Cys282Tyr in the British Isles and Ireland. They look at diagnostic rates and morbidity and propose areas of the region in which to focus screening. The study is very well conducted and provides important data for understanding the epidemiology of this disorder in the part of the world with the highest prevalence. Some issues that should be addressed are:

1. The correct nomenclature is p.Cys282Tyr. p.C282Y should no longer be used.
2. Line 47- "Symptoms range from hyperpigmentation..." Hyperpigmentation is a symptom of very severe iron overload and therefore should come in the latter part of the sentence with liver fibrosis and cirrhosis rather than in list of symptoms that are non-specific and generally related to lower iron overload.
3. Line 64- change "This was an important step change.." to "This was an important step..."
4. How representative of the UK population is the UK Biobank across the many different regions assessed in this study?
5. Line 192- p.C282T should be p.C282Y which in turn should be p.Cys282Tyr!
6. Line 220- p.H63A should be p.H63D which in turn should be p.His63Asp.
7. Line 223- variants in HJV and HAMP are associated with juvenile haemochromatosis. The other genes are not generally associated with that phenotype.
8. Line 370- p.Cys282Tyr homozygotes should be referred to as such rather than as "carriers" which is a term that should be used for heterozygotes.
9. The suggestion that different segments of a region as small as the British Isles and Ireland should have different approaches to screening for haemochromatosis is unwise, and in fact unethical, in my view. A person with the average risk of a white person from England who lives in a low prevalence area should have the same access to screening as a person of Irish ancestry living in Ireland. In addition, it is impractical to offer screening programs to subsets of a population based on geographical location. I suggest the discussion be revised to at least note these issues.

Reviewer #2

(Remarks to the Author)

This manuscript describes rates of genetic risk for hereditary hemochromatosis and clinical diagnoses of hereditary hemochromatosis across geographic regions in the British Isles and Ireland. It also summarizes rates of genetic risk across ancestral groups. The manuscript, which is generally clearly written, highlights high, but variable, prevalence of genetic risk across regions and potential under-ascertainment of at-risk individuals in certain regions. The methods are thoughtful, clearly described, and appropriate for the study's goals. Findings could have implications for genomic screening for hereditary hemochromatosis, though additional clarity on the points below could inform how significant these findings are for the design and anticipated outcomes of population genomic screening programs.

1. Given that the rate of individuals with the major risk HFE genotype is relatively high across all regions and that, as the authors point out, genomic screening that includes the major risk HFE genotype is believed to be cost-effective, one could argue that findings provide further support for genomic screening for hereditary hemochromatosis across all regions in the study, rather than for targeting screening to certain regions. Yet, given the exceedingly low rates of individuals with the major risk genotype in some ancestral groups, this "screen everyone" approach might not be optimal. Beyond suggesting that

genomic screening be targeted to regions that appear to under-ascertain at-risk individuals, what do study findings suggest about how genomic screening programs in ancestrally diverse populations should consider inclusion of the major HFE risk genotype?

2. Consider shortening the Discussion in ways that minimize restatement of results and focus on interpretation of findings.
3. It is at times difficult to follow exactly what genetic risk is being discussed. I would suggest clearly defining each related term (genetic risk, major risk genotype, combined genetic risk) early in the manuscript and using them consistently throughout. It should be clear when, for example, p.H63D/p.C282Y compound heterozygotes are included with p.C282Y homozygotes in calculations and when genetic risk is multiplied by the cumulative incidence of hemochromatosis.
4. If possible, it would be helpful to report the sensitivity and specificity of the E83.1 ICD-10 code for hemochromatosis. While the code appears to be commonly applied in individuals with HFE variants, it is not clear that it indicates iron overload (and may just indicate the presence of genetic risk in individuals with known HFE variants).
5. In the Introduction, consider expounding upon what is known regarding the association between early detection and treatment of hemochromatosis and prevention or reversal of end organ damage.

A few minor suggestions:

6. Please include citation(s) for the statement in line 51 regarding low penetrance of pathogenic HFE variants.
7. Please include citation(s) for the statement in lines 93-94 that targeting population genomic screening to areas with greater numbers of p.C282Y homozygotes would improve cost-effectiveness.
8. Consider moving the description of methods for assessing frequencies of non-HFE variants (lines 223-228) to the Methods section.
9. Consider explaining why hemochromatosis prevalence data from the NHS are not available for Scotland, Wales and Northern Ireland (lines 346-347).
10. My understanding is that ACMG prefers the phrasing 'ACMG secondary findings v3.0' list (e.g., in line 365).
11. Our Future Health is first mentioned in line 371 – I suggest mentioning earlier if relevant or explaining here.

Version 1:

Reviewer comments:

Reviewer #1

(Remarks to the Author)

I am happy with the responses to the reviews except for the response to the issue of differential screening based on geographic location. The example given where a Jewish person in England can have screening for BRCA but a person in Scotland with the same risk not having access is not a justification for differential screening for haemochromatosis. It is clearly unethical that in the same country that different rules apply based on geographic location. I believe the authors should be making this point rather than suggesting that a person with high risk ancestry living in a lower prevalence region should not have access to screening available to a person with the same ancestry living in a higher prevalence region.

Reviewer #2

(Remarks to the Author)

This is a thoughtful revision that improves clarity and context of the methods and findings. I have no further comments or suggestions.

Response to review Kerr et al. The landscape of hereditary haemochromatosis risk and diagnosis across the British Isles and Ireland

Reviewer #1:

Kerr and colleagues present data on the prevalence of HFE p.Cys282Tyr in the British Isles and Ireland. They look at diagnostic rates and morbidity and propose areas of the region in which to focus screening. The study is very well conducted and provides important data for understanding the epidemiology of this disorder in the part of the world with the highest prevalence.

We thank the reviewer for these kind words.

Some issues that should be addressed are:

1. The correct nomenclature is p.Cys282Tyr. p.C282Y should no longer be used.

We have now altered all instances of p.C282Y to p.Cys282Tyr.

2. Line 47- "Symptoms range from hyperpigmentation..." Hyperpigmentation is a symptom of very severe iron overload and therefore should come in the latter part of the sentence with liver fibrosis and cirrhosis rather than in list of symptoms that are non-specific and generally related to lower iron overload.

We thank the reviewer for this information and have now re-arranged the sentence such that hyperpigmentation is adjacent to the other symptoms related to more severe iron overload.

3. Line 64- change "This was an important step change.." to "This was an important step..."

We have deleted the word "change."

4. How representative of the UK population is the UK Biobank across the many different regions assessed in this study?

While it is well recognised that UK Biobank, along with all cohorts which involve volunteering, does not accurately represent the UK population socioeconomically, we do not believe that these biases are likely to have operated differentially across different parts of the UK. We make this point around lines 404-407:

"We recognise that no volunteer cohort can be a perfect representation of the population from which it is drawn. There is evidence of a "healthy volunteer" selection bias in the UKB³⁸, but we consider it unlikely that these biases operated to different degrees in different parts of the UK, such that they would substantively affect our conclusions."

5. Line 192- p.C282T should be p.C282Y which in turn should be p.Cys282Tyr!

We thank the reviewer for catching this typo, which we have now fixed to the correct notation.

6. Line 220- p.H63A should be p.H63D which in turn should be p.His63Asp.

We thank the reviewer for picking up on this mistake, which we have now fixed, along with all instances of p.H63D.

7. Line 223- variants in HJV and HAMP are associated with juvenile haemochromatosis. The other genes are not generally associated with that phenotype.

We have now removed mention of “mostly juvenile” so as not to imply any of the other genes mentioned are involved in the risk of this form of haemochromatosis.

8. Line 370- p.Cys282Tyr homozygotes should be referred to as such rather than as “carriers” which is a term that should be used for heterozygotes.

We thank the reviewer for this advice and have now explicitly named p.Cys282Tyr homozygotes as such. There are no other places in the paper where carrier has been used to refer to a homozygote.

9. The suggestion that different segments of a region as small as the British Isles and Ireland should have different approaches to screening for haemochromatosis is unwise, and in fact unethical, in my view. A person with the average risk of a white person from England who lives in a low prevalence area should have the same access to screening as a person of Irish ancestry living in Ireland. In addition, it is impractical to offer screening programs to subsets of a population based on geographical location. I suggest the discussion be revised to at least note these issues.

We thank the reviewer for sharing their opinion on this matter. We believe that screening should be available to all, but we doubt that this is going to be possible at least in the beginning, given the resource implications. We have, however, noted this more clearly by modifying a sentence in the discussion (around line 4810 to read as follows:

“While we advocate that screening should be available to all, given the resource implications, the data presented here suggest prioritisation of populations for screening in a variety of ways.”

In this connection, we note that screening is already available to subsets of the UK population, for instance, breast, ovarian and prostate cancer risk screening of the *BRCA1* and *BRCA2* genes is available for individuals with one or more Jewish grandparents, and who are under the care of NHS England (but not NHS Scotland and not for people without Jewish grandparents):

<https://nhsjewishbrcaprogramme.org.uk/>

Moreover, Jnetics provides screening for 47 recessive Mendelian disorders in the Jewish population in the UK:

<https://www.jnetics.org/>

Reviewer #2:

This manuscript describes rates of genetic risk for hereditary hemochromatosis and clinical diagnoses of hereditary hemochromatosis across geographic regions in the British Isles and Ireland. It also summarizes rates of genetic risk across ancestral groups. The manuscript, which is generally clearly written, highlights high, but variable, prevalence of genetic risk across regions and potential under-ascertainment of at-risk individuals in certain regions. The methods are thoughtful, clearly described, and appropriate for the study’s goals. Findings could have implications for genomic screening for hereditary hemochromatosis, though additional clarity on the points below could inform how significant these findings are for the design and anticipated outcomes of population genomic screening programs.

We thank the reviewer for this summary.

1. Given that the rate of individuals with the major risk HFE genotype is relatively high across all regions and that, as the authors point out, genomic screening that includes the major risk HFE

genotype is believed to be cost-effective, one could argue that findings provide further support for genomic screening for hereditary hemochromatosis across all regions in the study, rather than for targeting screening to certain regions. Yet, given the exceedingly low rates of individuals with the major risk genotype in some ancestral groups, this “screen everyone” approach might not be optimal. Beyond suggesting that genomic screening be targeted to regions that appear to under-ascertain at-risk individuals, what do study findings suggest about how genomic screening programs in ancestrally diverse populations should consider inclusion of the major HFE risk genotype?

We agree that the findings provide further support for screening across all regions, and we have now stated that in the discussion (lines 481-483):

“While we advocate that screening should be available to all, given the resource implications, the data presented here suggest prioritisation of populations for screening in a variety of ways.”

We go on to suggest a number of options for prioritisation.

2. Consider shortening the Discussion in ways that minimize restatement of results and focus on interpretation of findings.

We have now reduced the restatement of results in the discussion by deleting various sentences or parts of sentences around lines 395-398.

3. It is at times difficult to follow exactly what genetic risk is being discussed. I would suggest clearly defining each related term (genetic risk, major risk genotype, combined genetic risk) early in the manuscript and using them consistently throughout. It should be clear when, for example, p.H63D/p.C282Y compound heterozygotes are included with p.C282Y homozygotes in calculations and when genetic risk is multiplied by the cumulative incidence of hemochromatosis.

We thank the reviewer for this comment. We have now defined “major risk genotype” in the third line of the introduction. Combined genetic risk is defined at lines 290-293. Throughout the manuscript we have now been very careful to distinguish overall or combined genetic risk and major risk genotype.

4. If possible, it would be helpful to report the sensitivity and specificity of the E83.1 ICD-10 code for hemochromatosis. While the code appears to be commonly applied in individuals with HFE variants, it is not clear that it indicates iron overload (and may just indicate the presence of genetic risk in individuals with known HFE variants).

We thank the reviewer for this comment. Lucas *et al.*, 2024 have investigated this and we report their figures in the manuscript (lines 86-92) – approximately 90% of people with the E83.1 code have a genetic risk for haemochromatosis arising from p.Cys282Tyr or p.His63Asp. However, it is true that in some cases this may only be an indication of the presence of risk, rather than of symptoms related to iron overload, and we now make this explicit in lines 96-97:

“but we note it is possible that in some cases the code only indicates the presence of the major risk genotype, rather than of iron overload.”

5. In the Introduction, consider expounding upon what is known regarding the association between early detection and treatment of hemochromatosis and prevention or reversal of end organ damage.

We thank the reviewer for this suggestion and have now cited Niederau C. et al., (1996) *Gastroenterology* 110: 1107-19 on the fact that early diagnosis and treatment largely prevent the

adverse consequences of iron overload.

A few minor suggestions:

6. Please include citation(s) for the statement in line 51 regarding low penetrance of pathogenic HFE variants.

We thank the reviewer and have now added a citation for this statement.

7. Please include citation(s) for the statement in lines 93-94 that targeting population genomic screening to areas with greater numbers of p.C282Y homozygotes would improve cost-effectiveness.

This statement is made because it's logically impossible not to be the case. If cost effectiveness is measured by how many homozygotes one finds for a given amount of money spent, then it follows that the places with the highest frequency must be the most cost-effective places to screen. There is identical reasoning behind any screening programme that is targeted - e.g. to a particular age-group (such as bowel or breast cancer screening in the UK), ethnicity (e.g. Jewish BRCA screening) or cases of a disease (e.g. retinopathy screening for diabetics), indeed any subgroup that is at higher risk than the general population. Hence, cost-effectiveness must improve if screening, which has a fixed cost per person screened, is targeted to places with higher frequencies of the causal risk factor, in this case the p.Cys282Tyr genotype. Nevertheless, at lines 98-99 we have modified the sentence (altered text underlined) to read, "It seems clear that targeting to areas with greater numbers ... should improve cost-effectiveness."

8. Consider moving the description of methods for assessing frequencies of non-HFE variants (lines 223-228) to the Methods section.

We have moved two sentences from the section highlighted by the reviewer into the Methods.

9. Consider explaining why hemochromatosis prevalence data from the NHS are not available for Scotland, Wales and Northern Ireland (lines 346-347).

We wish that aggregated haemochromatosis (and other) prevalence data were available from Scotland, Wales and Northern Ireland, but no joined-up data service equivalent to DigiTrials exists in these nations. For example, in NHS Scotland, similar data would involve applications to 14 different health boards, which was not possible with the funding (and therefore time) available. However, we believe that discussion of the data limitations outside NHS England DigiTrials is beyond the scope of our paper.

10. My understanding is that ACMG prefers the phrasing 'ACMG secondary findings v3.0' list (e.g., in line 365).

We thank the reviewer for noticing this error, which we have now fixed by using the preferred phrasing.

11. Our Future Health is first mentioned in line 371 – I suggest mentioning earlier if relevant or explaining here.

We thank the reviewer for pointing this out. We have now deleted the mention of Our Future Health, as it was extraneous.

Response to second round of review

Reviewer #1:

I am happy with the responses to the reviews except for the response to the issue of differential screening based on geographic location. The example given where a Jewish person in England can have screening for BRCA but a person in Scotland with the same risk not having access is not a justification for differential screening for haemochromatosis. It is clearly unethical that in the same country that different rules apply based on geographic location. I believe the authors should be making this point rather than suggesting that a person with high risk ancestry living in a lower prevalence region should not have access to screening available to a person with the same ancestry living in a higher prevalence region.

Of course, we believe that the Northern Irish and Hebridean diasporas (in England or Mainland Scotland, for example) should be screened, but in most cases, the screening tends to begin where the at-risk population is more concentrated, which is where the cost-benefit is going to be most pronounced. This may not be ideal, but it is likely to be the reality, particularly at the start of a screening programme, i.e., before its effectiveness for a full rollout can be fully assessed. It is likely that this approach would also be a powerful approach to raise awareness in the descendant communities.

To temper our proposals, we have modified the wording around targeted screening, e.g. in the abstract, instead of proposing “priority areas in which to offer population screening”, we now “suggest health economic modelling of community screening should be targeted to these priority areas.”

In the introduction, we say: “to allow any work to build the evidence base for future screening programmes to prioritise areas of particularly high frequency, or discrepancies between overall genetic risk and prevalence.”

In the discussion, we have modified our sentence, beginning “While we advocate that screening should be available to all, given the resource implications, the data presented here suggest prioritisation of” to end “consideration of communities for screening, for example, through health-economic analyses, in a variety of ways.”

Reviewer #2:

This is a thoughtful revision that improves clarity and context of the methods and findings. I have no further comments or suggestions.

We thank the reviewer for their comments.